# Molecular Genetic Basis of Reproductive Fitness in Tibetan Sheep on the Qinghai-Tibet Plateau

**DOI:** 10.3390/genes16080909

**Published:** 2025-07-29

**Authors:** Wangshan Zheng, Siyu Ge, Zehui Zhang, Ying Li, Yuxing Li, Yan Leng, Yiming Wang, Xiaohu Kang, Xinrong Wang

**Affiliations:** 1School of Biological and Pharmaceutical Engineering, Lanzhou Jiaotong University, Lanzhou 730070, China; gesiyu0710@163.com (S.G.); liying_23@mail.lzjtu.cn (Y.L.); liyx@mail.lzjtu.cn (Y.L.); lengyan@mail.lzjtu.cn (Y.L.); wym030210@163.com (Y.W.); xhkang@mail.lzjtu.cn (X.K.); 2Provincial R&D Institute of Ruminants in Gansu, Lanzhou 730070, China; 3State Key Laboratory of Primate Biomedical Research, Kunming University of Science and Technology, Kunming 650500, China; kustzh@163.com; 4College of Animal Science and Technology, Gansu Agricultural University, Lanzhou 730070, China

**Keywords:** Tibetan sheep, reproductive success, nature selection, *HBB*, high-altitude adaptation

## Abstract

**Background**: Complete environmental adaptation requires both survival and reproductive success. The hypoxic Qinghai-Tibet Plateau (>3000 m) challenges reproduction in indigenous species. Tibetan sheep, a key plateau-adapted breed, possess remarkable hypoxic tolerance, yet the genetic basis of their reproductive success remains poorly understood. **Methods**: We integrated transcriptomic and genomic data from Tibetan sheep and two lowland breeds (Small-tailed Han sheep and Hu sheep) to identify Tibetan sheep reproduction-associated genes (TSRGs). **Results**: We identified 165 TSRGs: four genes were differentially expressed (DEGs) versus Small-tailed Han sheep, 77 DEGs versus Hu sheep were found, and 73 genes were annotated in reproductive pathways. Functional analyses revealed enrichment for spermatogenesis, embryonic development, and transcriptional regulation. Notably, three top-ranked selection signals (*VEPH1*, *HBB*, and *MEIKIN*) showed differential expression. Murine Gene Informatics (MGI) confirmed that knockout orthologs exhibit significant phenotypes including male infertility, abnormal meiosis (male/female), oligozoospermia, and reduced neonatal weight. **Conclusions**: Tibetan sheep utilize an evolved suite of genes underpinning gametogenesis and embryogenesis under chronic hypoxia, ensuring high reproductive fitness—a vital component of their adaptation to plateaus. These genes provide valuable genetic markers for the selection, breeding, and conservation of Tibetan sheep as a critical genetic resource.

## 1. Introduction

Environmental factors drive evolutionary processes through selective pressures on fecundity (quantified by viable offspring count) and mortality (measured as reproductive-stage survival rates) [1,2]. This selective process determines reproductive fitness—an organism’s capacity to successfully propagate its genetic material in specific environments [3,4]. Darwinian adaptation requires the concurrent optimization of a survival advantage and reproductive success [5]. Notably, diminished reproductive capacity manifests as reduced conception rates and an attenuated breeding efficiency, with sexual dimorphism exhibited in mammalian species [2].

The hypoxic environment of the Qinghai-Tibetan Plateau imposes significant selective constraints on human and domesticated animal reproduction [3]. Chronic high-altitude hypoxia (HAH) induces intrauterine growth restriction, leading to decreased birth weights (BW) and elevated perinatal mortality [6,7]. Through long-term natural selection, indigenous populations [8] and livestock species [9,10,11] have evolved adaptive reproductive strategies characterized by increased neonatal mass and reduced prenatal/postpartum mortality [3,7,12]. The Tibetan sheep (*Ovis aries*), one of China’s three primitive ovine breeds endemic to altitudes exceeding 3000 m, exemplifies exceptional hypoxic adaptation through evolutionary specialization [9,13,14]. Molecular ecological studies reveal the precise synchronization of Tibetan sheep reproductive cycles and alpine growing seasons, mediated by the photoperiod-responsive regulation of gonadotropin-releasing hormone (GnRH) secretion through the hypothalamic–pituitary–gonadal (HPG) axis [15,16]. Multi-omics evidence demonstrates enhanced oxygen utilization systems in their reproductive organs, with sustained HIF-1α signaling activation maintaining follicular development under hypoxia [16,17]. Evolutionary genetic analyses identify positive selection signatures in *EPAS1* (endothelial PAS domain protein 1), where adaptive SNPs enhance placental angiogenesis and associate strongly with increased lamb birth weight [18]. Seasonal reproductive gene expression is further modulated by DNA methylation dynamics [19].

In the past few decades, many studies have elucidated Tibetan sheep’s survival adaptations at physiological and genomic levels [9,14,15,17,18,20]. However, critical gaps remain in our understanding of the genetic architecture underlying their reproductive success. Therefore, we employ an integrative genomics approach, combining transcriptomic and whole-genome sequencing data from altitudinally stratified Tibetan sheep populations and lowland migratory breeds (Small-tailed Han and Hu sheep). This comparative framework enables the systematic identification of selection signatures and hypoxia-responsive regulatory networks (e.g., HIF pathways) governing reproductive fitness, providing mechanistic insights for high-altitude adaptation and precision breeding strategies.

## 2. Results

### 2.1. Sampling, Transcriptomic and Genomic Sequencing

We collected 22 testicular and 12 ovarian tissue samples from high-altitude-adapted sheep (*Ovis aries*), including Tibetan sheep (ten testicular samples and six ovarian samples), Small-tailed Han sheep (six testicular samples), and Hu sheep (six testicular samples and six ovarian samples), for subsequent transcriptomic and genomic sequencing. To broaden our genomic context, these newly generated data were integrated with publicly available whole-genome sequences from 162 Tibetan sheep and 30 lowland sheep [13,14,15,18,20] (Figure 1A; Appendix A). For the testicular transcriptome analysis, comparative RNA-seq was performed using lowland-adapted Small-tailed Han and Hu sheep that had migrated to plateau environments as controls. This approach identified 79 Tibetan sheep-vs-Small-tailed Han differentially expressed genes (TSHDEGs) and 310 Tibetan sheep-vs-Hu DEGs (THDEGs) (Figure 1B, Appendix A). Gene Ontology (GO) and KEGG pathway analyses revealed significant enrichment of TSHDEGs in flagellated sperm motility pathways (adjusted *p* < 0.05), which aligns with prior reports on high-altitude reproductive adaptations (Figure 1D) [21].

For ovarian transcriptome sequencing, we identified 1459 Tibetan sheep-vs-Hu differentially expressed genes (THODEGs) (adjusted *p* < 0.01; Figure 1C, Appendix A; see Section 4, Methods). The GO and KEGG pathway enrichment analyses revealed that the THODEGs were significantly enriched (adjusted *p* < 0.05) in DNA damage response and regulation of mRNA stability (Figure 1F, Appendix A). Strikingly, RNA-level regulatory mechanisms represent an evolutionarily conserved adaptive strategy for maintaining high reproductive fitness under chronic hypoxia; this molecular convergence aligns with transcriptomic profiles observed in the placentas of Tibetan highlanders [7]. Notably, the number of THODEGs was several-fold higher than that of TSHDEGs and THDEGs, highlighting the greater complexity of ovarian gene regulation under these conditions. Furthermore, we observed divergent expression patterns for three genes between tissues. For example, *HBB* expression was significantly downregulated in testes but significantly upregulated in ovaries (Figure 1B,C, Appendix A), suggesting potential sex-specific differences in the regulation of high-altitude fitness traits in Tibetan sheep.

To elucidate the evolutionary significance of TSHDEGs, THDEGs, and THODEGs, we performed cross-species genomic alignment against the human and mouse reference genomes, identifying 73 TSHDEGs, 256 THDEGs, and 1349 THODEGs as evolutionarily conserved orthologs. Integrated multi-omics analysis (KEGG, GO, and Mouse Genome Informatics) showed the enrichment of these conserved orthologs predominantly in reproductive process regulation, sperm motility, and spermatid development (Figure 1E). Importantly, these conserved functional signatures corresponded with ovine-specific enrichment patterns, suggesting that TSDEG-mediated optimization of testicular development and spermatogenesis may enhance male Tibetan sheep fertility, thereby maintaining reproductive fitness under hypoxic stress. In contrast, THDEGs exhibited distinct enrichment profiles associated with cellular homeostasis, developmental regulation, stress response, and reproductive processes (Appendix A). This multi-layered regulatory architecture implies potential compensatory mechanisms for coordinating testicular function with environmental challenges. Conversely, THODEGs were significantly enriched in pathways related to organelle localization and red blood cell count, which is consistent with the earlier observation of different regulatory patterns between ovarian and testicular tissues (Appendix A).

Cross-referencing our DEG sets against 1419 published reproduction pathway genes revealed 112 overlapping DEGs (13 THDEGs, five TSHDEGs, and 94 THODEGs) (Appendix A). This overlap not only reinforces their putative roles in high-altitude fitness but also further substantiates their adaptive significance in plateau environments.

### 2.2. Genome-Wide Natural Selection Signatures Scanned in Tibetan Sheep

Tibetan sheep maintain good reproductive fitness under hypoxic plateau conditions, and this phenotype has underlying genetic determinants. To investigate whether DEGs reflect genomic adaptations shaped by natural selection, we performed a genome-wide scan using the composite of multiple signals (CMS) method [23] (see Section 4). Finally, we designated SNVs within the top 0.5% of CMS scores (CMS > 7.06) as Tibetan sheep natural selection SNVs (TSNSSs). This approach identified 126,075 TSNSSs, spanning 588 genomic regions. Of these, 59 regions lacked gene annotations, while 529 overlapped with Tibetan sheep natural selection genes (TSNSGs) (Figure 2A, Appendix A). Notably, 472 TSNSGs represent novel identified genes, whereas 57 have been previously reported.

For the functional annotation of the 126,075 TSNSSs, we utilized SnpEff [24]. The results revealed that the majority of the TSNSSs reside in non-coding regions (specifically, 39.7% in introns and 56.6% in intergenic regions) (Figure 2B, Appendix A). Of particular interest, we detected 845 protein-truncating variants (PTVs), 493 missense mutations, eight stop-loss, four start-loss, 245 start-codon gain, and 95 splice-region variants (Appendix A). The highest-scoring PTV of 15:48037294 (CMS = 13.97; PBS = 0.37; XPEHH = 7.72; iHS = 4.11) occurred at Chr15:48,037,294 within *HBB*—a pivotal hemoglobin gene. This finding suggests the mutation contributes to hemoglobin regulation and consequently oxygen transport/dissociation in Tibetan sheep.

To functionally characterize the TSNSGs, we further ortholog-mapped TSNSGs to human and mouse genomes. Twenty-four genes exhibited a strong hypoxia response element (HRE) signature, 19 genes acted as core regulators in the hypoxia-inducible factor (HIF) pathway (e.g., *EP300*), and 34 genes had been previously associated with Tibetan high-altitude adaptation (Appendix A). Leveraging the well-curated databases of these model organisms, we observed significant enrichment (*p* < 0.01) in terms including embryo development ending in birth or egg hatching, tube morphogenesis, hemopoiesis, and skeletal system development (Appendix A). This developmental and hematopoietic convergence aligns with the reproductive system functions annotated in sheep databases, reinforcing the conclusion that TSNSGs mediate high-altitude adaptation through coordinated improvements in oxygen delivery (i.e., hemopoietic systems) and reproductive efficiency.

Remarkably, seven of the top ten TSNSGs constitute novel identified genes (Appendix A). Consistent with our GO results, Mouse Genome Informatics (MGI) annotations reveal that the knockout (heterozygous or homozygous) of these seven orthologous genes disrupts hematopoiesis, cardiopulmonary function, and/or reproduction (Table 1, Appendix A). This multisystem phenotypic coherence strongly supports the hypothesis that Tibetan sheep adaptation involves integrated genomic regulation across physiological systems.

### 2.3. Identification of Genes Underpinning Reproductive Fitness in Tibetan Sheep

To further delineate genes associated with reproduction in Tibetan sheep, we integrated differentially expressed genes (DEGs) from reproductive tissues with genes implicated in reproductive processes (Appendix A). Specifically, we identified 165 DEGs subject to natural selection in testes and ovaries (four testis-specific, 15 testis-enriched, and 77 ovary-enriched), alongside 73 key reproductive pathway genes (Figure 2C, Appendix A). Given that hypoxia presents a major challenge to reproduction [3,8], we propose that Tibetan sheep harbor a suite of positively selected genes regulating their reproductive capacity to ensure their fitness under hypoxia.

Among these reproduction-associated genes, we highlight three strongly positive-selected DEGs: *VEPH1*, *HBB*, and *MEIKIN* (Figure 3). Ventricular Zone Expressed PH Domain Containing 1 (*VEPH1*), which is predicted to possess phosphatidylinositol-5-phosphate binding activity, is involved in the negative regulation of SMAD protein signal transduction and TGF-β receptor signaling pathways, with predicted localization to the endomembrane system. Significantly, *VEPH1* is characterized by multiple TSNSSs, with its peakSNV selection signal at chr1:230822774 (CMS = 14.98). This region exhibits pronounced haplotype decay (mean iHS = 4.58; mean XPEHH = 3.8) relative to low-altitude breeds (e.g., Small-Tailed Han and Hu sheep), indicating strong positive selection during high-altitude colonization (Figure 3A). Furthermore, there are two protein-truncating variants in *VEPH1* (1:230944111; CMS =7.17 and 1:230790961; CMS = 9.8) (Appendix A). Notably, convergent evidence from human GWAS links *VEPH1* SNVs to adult body height regulation [25], suggesting its likely contribution to growth regulation in Tibetan sheep.

*HBB* (a well-established hypoxia adaptation gene) primarily regulates hemoglobin oxygen-carrying capacity [27]. Multiple SNVs within the HBB gene region exhibited robust signatures of selection (PeakSNV: 15:48057063; CMS = 14.17; iHS = 3.82; XPEHH = 9.95) (Figure 3B, Appendix A). Furthermore, HBB was significantly enriched for multiple strongly selected missense mutations (Appendix A). This evidence strongly supports HBB as a key gene underlying high-altitude hypoxia adaptation in Tibetan sheep. Critically, fetal development in sheep occurs under profoundly hypoxic conditions. Supporting its reproductive role, *HBB* knockout murine models demonstrate that homozygous/heterozygous deletion directly compromises reproductive fitness, evidenced by reduced female fecundity and lower birth weights (as reported by MGI). Recent genomic analyses have revealed a high-frequency *HBB* haplotype (Haplotype A) in Tibetan sheep that confers superior oxygen kinetics through enhanced hemoglobin–oxygen affinity and dissociation efficiency [28]. This adaptation is driven by the tissue-specific upregulation of HBB in cardiopulmonary systems (heart and lungs) and skeletal muscle, thereby facilitating hypoxia tolerance at high altitudes [28]. Notably, the *HBB* locus is enriched with multiple high-frequency missense variants (CMS > 7.06; Appendix A). We propose that these mutations confer structural perturbations in the hemoglobin tetramer—potentially altering oxygen-binding dynamics and allosteric heme release—to enhance hypoxic adaptation. Critically, hypoxia-regulated *HBB* transcription in reproductive organs represents a key mechanism regulating embryonic development and spermatogenesis in this species.

Meiotic Kinetochore Factor (*MEIKIN*) participates in meiotic chromosome segregation and sister chromatid cohesion. 5:20429017 (CMS = 14.66), an intronic mutation within the *MEIKIN* gene, exhibits strong signatures of natural selection. Compared to migratory populations, it shows significant haplotype extension (iHS = 3.79, XPEHH = 4.39) as well as allele frequency enrichment (ΔDAF = 0.43) (Figure 3C, Appendix A). Human GWAS link *MEIKIN* to venous thromboembolism and basophil count [29], while murine knockout models exhibit pronounced male infertility, characterized by abnormal spermatogenesis, meiotic defects, and oligozoospermia (as reported by MGI). Additionally, several other strongly selected genes (e.g., *MITF* and *FSTL1*) (Appendix A) exhibit functional relevance to reproduction, notwithstanding the lack of detectable differential expression signals in our analysis.

In summary, reproductive fitness maintenance in Tibetan sheep operates through a polygenic regulatory basis. Tibetan sheep appear to have enriched a cohort of functionally coordinated reproduction-linked genes that, by regulating spermatogenesis and embryonic development under hypoxia, ensure reproductive success and sustain high fitness levels in this demanding environment.

## 3. Discussion

High-altitude hypoxia represents a significant challenge for human habitation and the survival of domestic livestock. Maintaining robust reproductive fitness is thus an essential prerequisite for permanent settlement in high-altitude regions. This study utilized low-altitude migrated sheep breeds (Small-tailed Han sheep and Hu sheep) as controls. By integrating transcriptomic and genomic data, we identified a core set of 165 genes associated with reproduction (comprising four TSHDEGs, 15THDEGs, 77 THODEGs, and 73 reproductive pathway genes). Subsequent functional annotation and enrichment analyses revealed that these genes are predominantly implicated in spermatogenesis and embryonic development, thereby providing genetic evidence to inform breeding strategies and germplasm conservation efforts for Tibetan sheep.

Building upon our previous research [21,30], we integrated existing sheep testicular transcriptome data and performed RNA sequencing on Tibetan sheep (ovarian tissue), using low-altitude breeds as controls. This approach led to the identification of 79 TSHDEGs, 310 THDEGs, and 1459 THODEGs (Appendix A). Consistent with our prior findings, functional enrichment analysis showed significant associations with spermatogenesis, embryonic development, and reproductive process. Notably, the THODEGs were significantly enriched in pathways related to DNA damage response and the regulation of mRNA stability (Figure 1F). Given that oogenesis and embryonic development involve precisely coordinated temporal and spatial gene expression alongside translational control, the regulation of transcript stability and DNA integrity occurs at critical control points. Intriguingly, a similar regulatory pattern is observed in placental transcriptomes during human high-altitude adaptation [7]. Furthermore, considering that elevated ultraviolet (UV) radiation is a prominent environmental stressor on the plateau [31], we hypothesize that these genes may facilitate adaptation by enhancing DNA damage repair mechanisms in response to high-altitude UV exposure. Collectively, the reproductive organ transcriptome data offer transcriptional-level insights into the mechanisms underpinning the high fitness of Tibetan sheep. A limitation of this study is the unavailability of ovarian tissue from age-matched, high-altitude-migrated Small-tailed Han sheep, resulting in a gap in comparative ovarian transcriptomics. Moreover, our samples were collected at the first point of sexual maturity, representing only a single timepoint in the reproductive process. Therefore, future studies will incorporate samples from multiple developmental stages, including placental tissue, to analyze the dynamic temporal expression profiles of key reproductive genes.

Employing the Composite of Multiple Signals (CMS) approach [23], we scanned the Tibetan sheep genome genome-wide for signatures of positive selection, identifying 529 Tibetan Sheep Naturally Selected Genes (TSNSGs), 57 of which were previously reported (Figure 2A, Appendix A). Functional annotation revealed enrichment for genes regulating hematopoiesis and skeletal system development. Critically, increased erythrocyte count and hemoglobin concentration are key phenotypes for hypoxic adaptation [3,8]. In alignment with this, we observed that 78 TSNSGs are involved in hematopoiesis, suggesting that Tibetan sheep may enhance oxygen delivery through hemoglobin regulation. Notably, among the top 10 most strongly selected TSNSGs, there are seven newly identified genes (Appendix A), four of which (*CALCR*, *MEIKIN*, *MYO3B*, and *FSTL1*) display murine knockout phenotypes significantly associated with hematopoiesis, cardiopulmonary function, and reproduction (Appendix A). Reinforcing the link to adaptation, the previously reported top hypoxia adaptation genes *HBB* and *MITF* (key regulators of hemoglobin synthesis) were also among the top TSNSGs, further supporting the role of these genes in adapting circulatory and reproductive systems to the plateau environment. It is pertinent to mention that established high-altitude adaptation genes *EPAS1* (CMS = 9.58) and *EGLN1* (CMS = 8.06) showed strong selection signals in Tibetan sheep but did not rank within the top 10. Studies indicate the presence of structural variations (SVs) in the *EPAS1* locus of Tibetan sheep [9,14], leading us to hypothesize that SVs, rather than single-nucleotide variants (SNVs), might be the primary targets of selection, with SNVs potentially exhibiting a hitchhiking effect. The absence of long-read sequencing data in this study precluded the direct validation of SVs in *EPAS1*, representing a key focus for future work. Furthermore, by restricting our analysis to sites within the top 0.5% genome-wide for CMS scores to define the TSNSG set, we acknowledge that genes under weaker selection may have been overlooked. Nevertheless, the identified set provides a robust foundation for elucidating the genetic mechanisms underlying high-altitude adaptation in Tibetan sheep.

Through the integration of DEGs from Tibetan sheep reproductive organs, key reproductive pathway genes, and the TSNSGs, we consolidated the identification of 165 genes exhibiting signatures of both natural selection and differential expression. These genes are significantly enriched for functions in spermatogenesis and embryonic development, a finding congruent with prior histomorphological studies [21,30,32]. A particularly striking observation was the highly significant differential expression of *HBB*, a master regulator of hemoglobin, in both testes and ovaries, yet with contrasting expression patterns between the sexes. This apparent discrepancy strongly indicates the existence of sex-specific molecular strategies for maintaining high reproductive fitness in Tibetan sheep.

In conclusion, this integrated analysis of transcriptomic and genomic data enabled the identification of 529 robust Tibetan Sheep Naturally Selected Genes (TSNSGs), 165 of which are implicated in regulating the reproductive system. *HBB*, *VEPH1*, and *MEIKIN* emerge as the most strongly selected DEGs with established reproductive functions. Functional evidence from murine models strongly links these genes to infertility (in both sexes), abnormal meiosis, and oligozoospermia (MGI data). Collectively, this work provides novel insights into the genetic architecture underpinning exceptional reproductive fitness in Tibetan sheep and offers crucial genetic support for selective breeding programs. Moving forward, the acquisition of multi-omics data from reproductive organs across different developmental stages, coupled with cellular and animal model validation of key candidate genes, will be paramount.

## 4. Data and Method

### 4.1. Sample Collection

Building upon published data [21,30], we collected gonadal tissues from 22 healthy sexually mature Tibetan sheep (9 females and 13 males) and 6 migratory Hu sheep (females), all aged 1.5 years at Haiyan County, Qinghai (elevation: 3500 m). All subjects met the following criteria: synchronized birth dates, identical feeding regimens, and body weight variations <4 kg. All studied ewes were maintained in the luteal phase during sampling/experiments. Notably, the heightened hypoxic stress at high altitudes imposes greater adaptive challenges on Small-tailed Han sheep (*Ovis aries*), restricting their local population size. This limitation precluded collection of samples from age-matched, luteal-phase-synchronized ewes with the same ovarian status at the same elevation during our study. Testicular/ovarian tissues from identical anatomical positions were dissected and divided into three aliquots: one for histological sectioning and another submitted to a commercial provider for RNA-seq (Illumina platform) and whole-genome sequencing (6 Tibetan sheep and 6 Hu sheep). In accordance with the Declaration of Helsinki, the protocol of this study was reviewed and approved by the Internal Review Board of Gansu Agricultural University (Approval ID: GSAU-Eth-AST-2025-001, 25 February 2025), and the research scheme is in accordance with the Regulations of the People’s Republic of China on the Administration of Human Genetic Resources.

Additionally, we collected published whole-genome data (WGS) of Tibetan sheep, Small-tailed Han sheep, and Hu sheep [9,14,18,28,32]. To expand the dataset, we newly generated whole-genome sequences from 12 individuals (6 Tibetan sheep and 6 Hu sheep) and integrated 192 publicly available genomes. This resulted in a final cohort of 204 samples (Appendix A). Genomic and transcriptomic datasets were integrated to decipher genetic mechanisms maintaining exceptional reproductive fitness in Tibetan sheep subjected to chronic high-altitude hypoxic stress. Both the transcriptomic and genomic samples were drawn from the same subspecies/clonal group.

### 4.2. Transcriptome Analysis

#### 4.2.1. Data Preprocessing

Using fastp (v0.23.2) [33], we performed quality control in the following steps: ① removal of low-quality reads (Q < 20); ② adapter trimming; ③ exclusion of reads with >5% N bases; ④ discarding reads < 150 bp.

#### 4.2.2. Differential Expression Analysis

Alignment: High-quality reads were aligned to the Ovine reference genome ARS-UI_Ramb_v3.0 (GCF_016772045.2) using STAR (v2.7.11b) [34] to generate BAM files.

Quantification: Sorted BAM files (samtools, v1.22) were processed with featureCounts (v2.0.3) [35] to generate individual gene count matrices that were subsequently merged into a unified expression matrix.

Differential Expression Analysis: Prior to DESeq2 (v1.42.0) analysis, genes with <10 reads across all samples were filtered.

#### 4.2.3. Functional Enrichment

KEGG/GO enrichment was conducted using clusterProfiler (v4.10.0) and AnnotationDbi (v1.62.0) with sheep annotation package “AH7526” for gene ID conversion. Significance thresholds were adjusted via Benjamini–Hochberg correction. Results were visualized with ggplot2 (v3.4.3). We employed org.Hs.eg.db (v3.21.0) and org.Mm.eg.db (v3.18.0) for cross-species function annotation. Furthermore, we consolidated data from multiple authoritative sources, including OMIM, Pathogenic Loss-of-Function variants (ClinVar), Genome-Wide Association Studies (GWAS) from NHGRI-EBI, Human Phenotype Ontology (HPO), and the Mouse Genome Informatics (MGI) database. To assess the enrichment significance of phenotypic traits, we employed Fisher’s exact test, with subsequent *p*-value adjustments made using the Bonferroni correction method to account for multiple comparisons.

### 4.3. Genomic Analysis

#### 4.3.1. Data Preprocessing

This process was identical to that for transcriptome QC (Section 2.1).

#### 4.3.2. Variant Calling

BWA-MEM (v0.7.17) was used to map reads to ARS-UI_Ramb_v3.0. Deduplication: PCR duplicates were marked/removed via GATK MarkDuplicates (v4.4.2). BaseRecalibrator was used to adjust base quality scores using Tibetan sheep INDEL sites.

Variant discovery: ① Individual gVCFs from HaplotypeCaller; ② Merged GVCFs via CombineGVCFs; and ③ Joint genotyping with GenotypeGVCFs.

### 4.4. Variant Quality Control

#### 4.4.1. Primary Filtering

Using GATK’s hard filters, the following was obtained: QD < 2.0 | QUAL < 30 | SOR > 3.0 | FS > 60.0 | MQ < 40.0 | MQRankSum < −12.5 | ReadPosRankSum < −8.0. This yielded 46,412,156 variants across 204 individuals.

#### 4.4.2. Sample-Level QC

Individuals with ≥3% missing genotypes or heterozygosity rates beyond mean ± 3SD were excluded. Eleven samples were removed (Appendix A).

Following LD pruning (PLINK v2.3), PCA was performed using EIGENSOFT (v7.2.1). Two samples were removed, and 191 individuals passed QC (Appendix A).

Thirteen samples were removed due to identity by descent (IBD).

#### 4.4.3. Variant-Level QC

VCFtools (v0.1.16) and plink2 (v2.1) were used to apply the following filters:①Removal of 2,808,163 singleton variants;②Exclusion of 4,712,207 SNVs with >5% missing genotypes;③Elimination of 983,635 SNVs violating Hardy–Weinberg equilibrium exact test (*p* < 1 × 10^−6^).

A total of 37,908,151 high-confidence SNVs were retained.

#### 4.4.4. Functional Annotation

Biallelic SNVs underwent allele frequency calculation. ANNOVAR (v2020-06-07) [36] and SnpEff (v5.1) [24] were used to annotate variants, with protein-truncating variants (PTVs) confirmed by both tools prioritized for downstream analysis.

### 4.5. Genome-Wide Natural Selection Signature Detection

Composite multiple signals (CMS) [23] scores were derived from four metrics (XP-EHH, iHS, PBS, and ΔDAF).

#### 4.5.1. Population Branch Statistic (PBS)

PBS was calculated for Tibetan sheep (T) relative to Hu (H) and Small-tailed Han (S) outgroups:PBS = (F*_ST_*_T-H_ + F*_ST_*_T-S_ − F*_ST_*_S-H_)/2

#### 4.5.2. ΔDAF

ΔDAF was derived as follows:DeltaDAF = ((DAF_T − DAF_S) + (DAF_T − DAF_H))/2
with DAF representing non-reference genomic mutation genotypes of Tibetan sheep (DAF_T) relative to Hu (DAF_H) and Small-tailed Han (DAF_S) sheep.

#### 4.5.3. XPEHH and iHS

Genome-wide scans used selscan (v1.3.1) [37], excluding 21,301 SNVs with EHH decay <0.05 prior to normalization.

#### 4.5.4. CMS Implementation

Following Deng [38], CMS scores integrated XPEHH, iHS, PBS, and ΔDAF values. Candidate loci (top 0.5%, n = 126,075) underwent LD-based clumping (PLINK parameters: r^2^ ≤ 0.2 and 500 kb window), identifying 529 independent selection regions. Tibetan natural selection genes (TNSGs) were defined as genes within ±5 kb of peak SNVs in each region, yielding 472 TNSGs.

## Figures and Tables

**Figure 1 genes-16-00909-f001:**
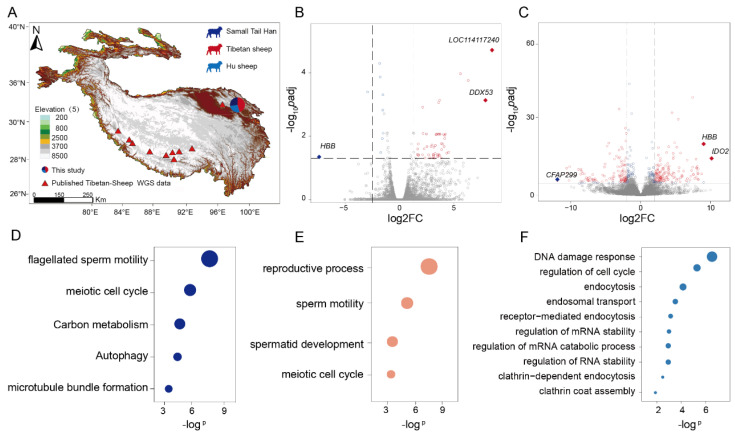
**Sample distribution and RNA-seq of Tibetan sheep**. (**A**) Geographical distribution of testicular samples from Tibetan sheep populations; (**B**) Differential expression analysis between Tibetan and Small-tailed Han sheep, identifying 92 TSHDEGs (labeled genes represent the top three most significant hits by log_2_FC); (**C**) Comparative transcriptomics of Tibetan vs. Hu sheep, revealing 235 THDEGs (top three log_2_FC genes annotated); (**D**) Functional enrichment of TSHDEGs using clusterProfiler (v4.10.0) and AnnotationDbi (v1.62.0) with sheep annotation package “AH7526”, with the results showing significant association with flagellated sperm motility pathways (adjusted *p* < 0.05; FDR is used for multiple *p* value testing correction); (**E**) Cross-species functional annotation (human/mouse references) of conserved TSHDEG orthologs generated by Metascape [22], highlighting enrichment in reproductive process regulation and spermatogenesis; (**F**) Enrichment profile of THDEGs in cellular homeostasis, stress adaptation, and reproductive pathways via clusterProfiler (v4.10.0) (annotation package “AH7526”).

**Figure 2 genes-16-00909-f002:**
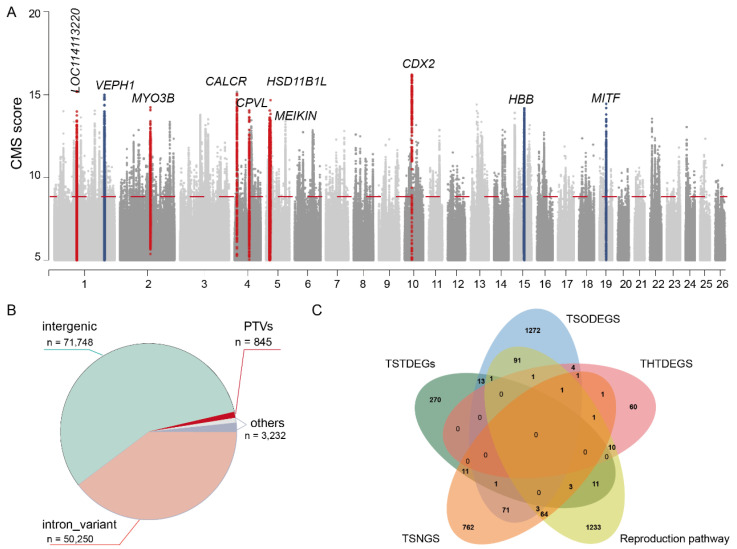
**Genome-wide scans for signatures of natural selection in Tibetan sheep.** (**A**). Manhattan plot depicting composite selection scores (CMS) for 25,327,494 single-nucleotide variants (SNVs). The red horizontal line (CMS = 7.06) indicates the top 0.5% threshold for significant selection signals. Novel candidate genes are highlighted in red, whereas previously reported hypoxia-adapted genes are shown in navy blue. The top 10 TSNSGs are annotated with gene symbols. (**B**). Pie chart categorizing genomic annotations of 126,075 trait-associated SNVs under selection (TSNSSs). The majority of SNVs reside in intergenic regions, while the “Others” category comprises 1126 upstream and 2106 downstream regulatory variants. (**C**). Tibetan sheep harbored 165 positive-selection reproduction-associated genes. This set includes 92 differentially expressed genes (DEGs) exhibiting selection signals in gonadal tissues—specifically, four testis-specific, 15 testis-enriched, and 73 ovary-enriched DEGs—alongside 73 key reproductive pathway genes subject to selection.

**Figure 3 genes-16-00909-f003:**
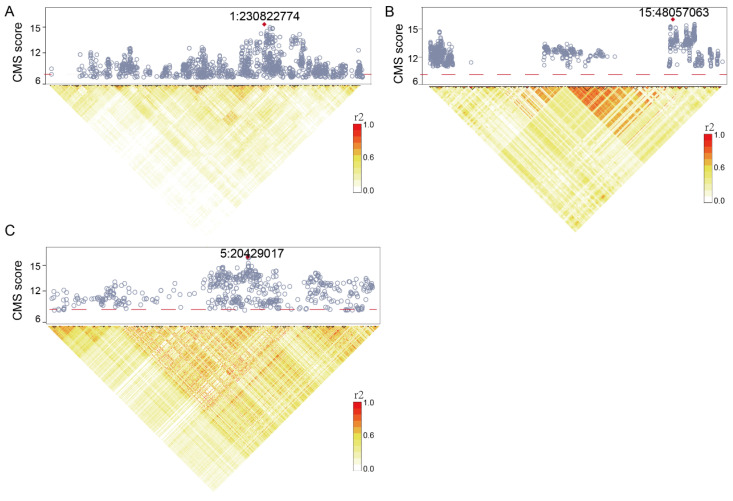
**Signatures of election in three reproduction-associated DEGs exhibiting high-altitude adaptation**. (**A**). *VEPH1* selection landscape. (Top) Scatter plot of CMS scores for SNVs across the *VEPH1* genomic interval (chr1:230.5–231.1 Mb). Peak selection signal (chr1:230822774; CMS = 14.98) is highlighted with a red diamond. (Bottom) Linkage disequilibrium (LD) heatmap of SNPs within *VEPH1*, with r^2^ values indicating pairwise correlation strength. Analogous plots to Panel (**A**) showing HBB (**B**,**C**) selection landscape. LDBlockShow [26] was used for LD visualization.

**Table 1 genes-16-00909-t001:** The natural selection information of top 10 TSNSGs.

CHR	SNVID	CMS	XPEHH	iHS	PBS	MDAF	Gene
10	10:32335294	16.82	3.91	2.91	0.41	0.42	*URAD*
1	1:103177691	15.28	3.55	2.40	0.37	0.43	*LOC114116898*
4	4:11649074	14.98	5.93	1.97	0.48	0.37	*CALCR*
1	1:230822774	14.98	3.80	4.52	0.23	0.33	*VEPH1*
5	5:20429017	14.66	4.38	3.70	0.14	0.43	*MEIKIN*
19	19:31783794	14.44	4.86	2.20	0.37	0.36	*MITF*
2	2:139020738	14.23	3.95	4.25	0.27	0.29	*MYO3B*
15	15:48057063	14.17	9.95	3.80	0.15	0.31	*HBB*
5	5:16451067	14.07	4.08	2.01	0.35	0.39	*HSD11B1L*
1	1:186064886	14.03	5.01	3.63	0.21	0.34	*FSTL1*

Note: CHR, chromosome; SNVID, the numbering information of SNV, which is composed of chromosomes and physical locations (ARS-UI_Ramb_v3.0); MDAF, the average value of the differences in allele frequencies of non-reference genomic genotypes between Tibetan sheep and Small-tailed Han sheep and Hu sheep.

## Data Availability

The whole-genome sequencing data utilized in this study have been deposited in the Genome Sequence Archive (GSA) under accession codes PRJCA007246, RJNA304478, PRJNA675420, PRJNA797957, PRJCA016837, PRJCA042340, and PRJNA624020. The variation data reported in this paper have been deposited in the Genome Variation Map (GVM) [1] in the National Genomics Data Center, Beijing Institute of Genomics, Chinese Academy of Sciences and China National Center for Bioinformation [39], under accession number GVM001099.

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
