# Peer review of "Molecular Genetic Basis of Reproductive Fitness in Tibetan Sheep on the Qinghai-Tibet Plateau"

_genes, 2025, doi:10.3390/genes16080909_

Round 1
Reviewer 1 Report
Comments and Suggestions for Authors
The manuscript in many cases sets the study’s context effectively. Data and method section is detailed and illustrate that the authors have a high level of expertise. However, some areas need improvement, such as clearer connection to a hypothesis or further clarification.
- I can understand authors’ need to demonstrate that they have performed a deep review of the literature, but, I found that text somehow is overwhelmed with the dense use of citations. The number of citations could be reduced following a critical approach without reducing the scientific robustness of the manuscript.
- The major drawback of the introduction to my opinion lies on the absence of setting a research question-hypothesis or objective.
- Lines 70-73. The sentence needs improvement, consider starting the sentence with a word like “Although” or use an alternative connecting word before “critical gaps”.
- Lines 73-79. The authors here should present their research question/hypothesis in view of the aforementioned literature and/or postulate their expectations. This will allow readers to be clear on the study’s focus.
- The term “hypoxia-responsive regulatory networks” needs some clarification to allow better understanding for readers outside molecular ecology or genomics.
- As mentioned before, the absence of a clear research question or hypothesis raises questions on the methodology used. The molecular approaches described in the “methods” section do not connect to a specific research question and it makes it harder for readers to understand how the methods directly address the study’s goals.
- Lines 356–366: Please provide details on the timing of tissue collection relative to reproductive cycles as this is important given the photoperiod-responsive GnRH regulation.
- Line 361. The term “unavailability of samples meeting equivalent growth conditions” is vague and needs further clarification.
- Lines 364, 370: There is an inconsistency on the numbers of samples analysed/reported. Line 364 states that WGS was performed on 10 Tibetan sheep and 6 Hu sheep, but line 370 mentions WGS data from 12 individuals (6 Tibetan sheep and 6 Hu sheep).
- Line 371. The cohort of 204 samples includes 192 public genomes, but the breakdown of species is unclear, please make it clear.
Author Response
We thank the reviewers for their constructive feedback on our manuscript and the editor for facilitating this process. All comments have been addressed point-by-point below, with corresponding revisions highlighted in the manuscript(see attachment ).

Reviewer 2 Report
Comments and Suggestions for Authors
Molecular Genetic Basis of Reproductive Fitness in Tibetan Sheep on the Qinghai-Tibet Plateau
Dear Authors,
The manuscript is interesting and well-prepared. The obtained results have valuable practical application for the conservation of Tibetan sheep genetic resources due to the increased reproductive success confirmed by the presence of three top-ranked selection signals during analysis. Additionally, they may constitute a valuable genetic resource for crossbreeding and improving the production performance of livestock animals at higher altitudes. The most problematic element from the perspective of this form of manuscript is the lack of Conclusions. The remaining suggestions are mainly related to text editing.
Below I added some suggestions helpful in revision process:
Line 16
Ovis aries, binominal nomenclature. Italics must be used.
Lines 14-20
Font must be adapted to the rest of lines in Abstract.
Line 41
Authors must be cited in manuscript in form of reference number/s, according to Instructions for Authors in order given in Reference section number. Fan et al., 2016; Moore 2017 can be emphasized as [1,2].
If Authors are emphasized in sentence, then instead of year of publication reference number must be given, i.e. Fan et al. [1].
Please check to the line 432.
Line 83
The same as in line 16.
Lines 134-146
Text must be justified, and font must be decreased about one unit.
Line 198
Table 1
In columns XPEHH and his two decimals must be present in case of each value.
Line 355
In test of manuscript as a title’s section is: 4. Data and Method, mut maybe better to describe as 4. Data and Methodology.
Style of sections 1-4 (and lacking 5) can be unified.
Line 452
Conclusions must be added.
Lines 471-547
References must be adapted to pattern described in Instructions for Authors.
https://www.mdpi.com/journal/genes/instructions
or even easier is to compare that in case of published articles, e.g.: https://doi.org/10.3390/genes16070776
I.e.:
- Astle, W.J.; Elding, H.; Jiang, T.; Allen, D.; Ruklisa, D.; Mann, A.L.; Mead, D.; Bouman, H.; Riveros-McKay, F.; Kostadima, M.A. et al. Cell 2016, 167(5), 1415-1429. http://dx.doi.org/10.1016/j.cell.2016.10.042
Author Response
We thank the reviewers for their constructive feedback on our manuscript and the editor for facilitating this process. All comments have been addressed point-by-point below, with corresponding revisions highlighted in the manuscript.

Reviewer 3 Report
Comments and Suggestions for Authors
The manuscript entitled “Molecular Genetic Basis of Reproductive Fitness in Tibetan Sheep on the Qinghai-Tibet Plateau” represents an interesting and, above all, innovative approach to the study of the reproductive success of the Tibetan sheep, an animal that has developed in its evolution an extraordinary resistance to hypoxia. Undoubtedly, the most interesting aspect is the integration of genomic and transcriptomic data that returns relevant answers to the initial question.
The conclusions reached by the Authors, namely that Tibetan sheep have a set of evolutionarily selected reproductively associated genes aimed at promoting gametogenesis and embryogenesis under conditions of chronic hypoxia, are of considerable merit.
While I have absolutely no complaints about the scientific approach to the subject, I found the introductory section of the manuscript and the discussion a tad lacking in information, especially in relation to comparisons with studies that have investigated in the same animals or related species other adaptive modes by considering genomics and transcriptomics. Several studies are available in the literature, even quite recent ones, which could be useful to the Authors precisely to enrich the introduction and discussion of the results that are moreover absolutely interesting and relevant.
Apart from these suggestions, which in my opinion could enrich the manuscript and determine an added value to the research.
Author Response

(The authors gave the same response as above.)

Round 2
Reviewer 2 Report
Comments and Suggestions for Authors
Dear Authors,
Thank you for revision process, but still some corrections must be done what was also described in previous report.
Below I added some suggestions helpful in revision process:
Line 17
Ovis aries, binominal nomenclature. Italics must be used.
Line 41
References are changed, but without spaces [1,2].
Please check to the line 438.
Line 79
The same as in line 17.
Line 191
Table 1
In columns XPEHH and iHS two decimals must be present in case of each value.
Line 345
In test of manuscript as a title’s section is: 4. Data and Method, mut maybe better to describe as 4. Data and Methodology.
Line 452
Conclusions must be added!!!
Lines 464-547
References must be adapted to pattern described available in Genes Instructions for Authors.
https://www.mdpi.com/journal/genes/instructions
or even easier is to compare that in case of published articles, e.g.: https://doi.org/10.3390/genes16070776
I.e.:
Astle, W.J.; Elding, H.; Jiang, T.; Allen, D.; Ruklisa, D.; Mann, A.L.; Mead, D.; Bouman, H.; Riveros-McKay, F.; Kostadima, M.A. et al. Cell 2016, 167(5), 1415-1429. http://dx.doi.org/10.1016/j.cell.2016.10.042
Author Response

(The authors gave the same response as above.)

Reviewer 3 Report
Comments and Suggestions for Authors
I believe that the authors have taken into account the observations made during the first revision of the manuscript.
Furthermore, they have adequately and satisfactorily responded to my concerns in the rebuttal letter.
Therefore, I believe that the manuscript can now be published without further changes.
Author Response
We thank the reviewers for their constructive feedback on our manuscript and the editor for facilitating this process.